**Data Availability Statement:** All relevant data are within the paper and its Supporting Information files.

**Funding:** The authors received no specific funding for this work.

# Perceived losses of scientific integrity under the Trump administration: A survey of federal scientists

**Gretchen T. Goldman**[1]*, **Jacob M. Carter**[1]*, **Yun Wang**[2], **Janice M. Larson**[2]

**1** Center for Science and Democracy, Union of Concerned Scientists, Washington, DC, United States of America, **2** Center for Survey Statistics and Methodology, Iowa State University, Ames, IA, United States of America

* ggoldman@ucsusa.org (GTG); jcarter@ucsusa.org (JMC)

## Abstract

President Trump and his administration have been regarded by news outlets and scholars as one of the most hostile administrations towards scientists and their work. However, no study to-date has empirically measured how federal scientists perceive the Trump administration with respect to their scientific work.

In 2018, we distributed a survey to over 63,000 federal scientists from 16 federal agencies to assess their perception of scientific integrity. Here we discuss the results of this survey for a subset of these agencies: Department of Interior (DOI) agencies (the US Fish and Wildlife Service (FWS), the US Geological Survey, and the National Park Service); the Centers for Disease Control and Prevention (CDC); the US Environmental Protection Agency (EPA); the Food and Drug Administration (FDA); and the National Oceanic and Atmospheric Administration (NOAA). We focus our analysis to 10 key questions fitting within three core categories that relate to perceptions of integrity in science. Additionally, we analyzed responses across agencies and compare responses in the 2018 survey to prior year surveys of federal scientists with similar survey questions. Our results indicate that federal scientists perceive losses of scientific integrity under the Trump Administration. Perceived loss of integrity in science was greater at the DOI and EPA where federal scientists ranked incompetent and untrustworthy leadership as top barriers to science-based decision-making, but this was not the case at the CDC, FDA, and NOAA where scientists positively associated leadership with scientific integrity. We also find that reports of political interference in scientific work and adverse work environments were higher at EPA and FWS in 2018 than in prior years. We did not find similar results at the CDC and FDA. These results suggest that leadership, positive work environments, and clear and comprehensive scientific integrity policies and infrastructure within agencies play important roles in how federal scientists perceive their agency's scientific integrity.

**Competing interests:** The authors have declared that no competing interests exist.

## Introduction

The use and misuse of science in policy development has spurred a long-standing discussion about the proper role of science and scientists in government decision-making. One important facet of this issue is the role of scientists who feed their knowledge and expertise into policy decision-making processes [1]. Scientists working for the federal government conduct, synthesize, and communicate scientific information that informs, guides, and directs policy action on issues with implications for public health and safety (e.g., the effects of climate change, the safety and efficacy of new drugs). This process has allowed government decisions to be better informed by scientific evidence producing great societal benefits [2–8].

The process by which science informs policy decisions is a complex "value-laden social process" and is affected by both scientists and non-scientist actors, as well as the environment in which it occurs [9, 10]. Thus, it is critical that postures (e.g., attitudes, stances, or perspectives) of all actors working at the science-policy interface are considered when they are contributing to policy decisions [11]. Such transparency can shield science-policy decisions from being overburdened by personal preferences that are presented as scientific fact and allow the public and other officials to observe and react to decisionmakers' actions [11–13]. In addition to transparency, scientific input from experts such as scientific advisory committees, academic scholars, or interdepartmental consultations, along with input from the public, can lead to robust and informed policy decisions. However, such expert-informed and participatory structures prove difficult in systems of government that have high concentrations of power in the executive branch, such as the US [14, 15]. In such a system, federal scientists and their work can be influenced by the values, culture, beliefs, and interests of those who hold more power in the U.S. bureaucracy. These elite actors may inappropriately undermine, alter, or otherwise interfere with the scientific process or scientific conclusions for political, financial, or ideological reasons. This is political interference in science, which often results in decisions more heavily influenced by political ideologies rather than scientific knowledge [16]. Political decisions that are not informed by science are not as likely to benefit society and, in some cases, may cause undue harm to the public.

Political interference has occurred in both Republican and Democratic administrations dating back as far as the Eisenhower administration [17]. However, political interference in science policy processes was first widely recognized under the George W. Bush administration [18]. The Union of Concerned Scientists, by its measure, counted nearly 100 incidents in which the administration sidelined scientists or their work when making science-based policy decisions [19]. The George W. Bush administration's political interference in the science-policy process prompted thousands of scientists to pen a letter entitled "Scientist Statement on Restoring Scientific Integrity to Federal Policymaking" to the administration [20]. This letter gave rise to a larger movement within the scientific community to push for integrity in processes at agencies where science informs decisions—referred to as "scientific integrity."

Ultimately, this push led to the development of scientific integrity policies across federal science agencies [21–23]. These policies are maintained and implemented by 28 federal entities that conduct and/or use science to inform decisions, and the provisions therein vary widely across agencies. The policies govern actions by staff who work at the science-policy interface and the provisions are designed to ensure scientific integrity is maintained and to provide a mechanism to report a violation of scientific integrity if it occurs. These provisions may include protections for scientists to communicate their work to the public without political interference, rights to submit work to an outside peer-reviewed journal, or the right of last review on agency statements that refer to a scientist's work.

While the US government science-policy apparatus is set-up to receive input on science-policy decisions from multiple stakeholders, ultimately science-policy decisions often lie in the hands of elite bureaucratic officials [24]. For example, the administrator of the Environmental Protection Agency (EPA) has a responsibility to consider scientific information provided from scientific advisory committees, stakeholder groups, the EPA's own scientific staff, and public opinion, but final decisions on the laws EPA is charged with carrying out (e.g. The Clean Air Act, the Safe Drinking Water Act, the Toxic Substances Control Act) often lie with the administrator. As mentioned previously, science policy is a value-laden social process and certainly the values of elite bureaucratic officials play a role in their science-policy decisions. If their decisions rely heavily on their own values and judgments and not on scientific information, this may lead to scientific work being deprioritized. The stifling of scientific work because it is viewed as politically contentious may demoralize scientific staff, make them feel ineffective, or result in self-censorship of their own work. Indeed, leadership, job effectiveness, and morale are often strongly correlated and can affect worker retention and production [25–27]. Research has indicated that an employee's leader has the most influence on their job satisfaction [27]. While research focusing on scientists in the workplace is scarce, existing research shows that they can be de-motivated by lack of support from their leaders [26, 27].

While few studies have assessed federal scientific integrity specifically, the US Office of Personnel Management conducts annual studies of federal employees, known as the Federal Employee Viewpoint Survey [28]. These surveys are inclusive of scientists and other technical experts in the federal workforce and assess employee satisfaction, engagement level, and allow for comparisons over time and between agencies. They have been used to assess the engagement of specific groups (e.g., women, millennials in the federal service) as well as specific challenges or events (e.g., impact of the 2019 federal government shutdown, education within the federal workforce) [29]. Some work has found correlations between the level of engagement found on OPM surveys and agency performance [30].

Additionally, workplace satisfaction surveys in technical fields have provided insights into scientists' job experiences. On its 2018 biannual survey of its readers with graduate scientific training, *Nature* found that 68% of the government scientists in its survey of 4,334 eligible respondents globally reported being satisfied with their job, and 21% of all respondents reported personally experiencing harassment or discrimination at their current job [31]. The American Association for the Advancement of Science has conducted similar surveys of its US-based technical readers and found an average job satisfaction rate of 3.7 on a five-point scale [32]. Both of these surveys were inclusive of US federal government scientists; however, results were not stratified by federal agency and scientific integrity at work was not a focus of the surveys. Other surveys have focused on the job satisfaction, job stress, and career trajectories of subpopulations and specific scientific disciplines, but respondent numbers and scope have been limited [33–34].

Some work has focused on job satisfaction and perceived ethics and fairness in the federal government context. Choi (2011) used the Merit System Protection Board Survey to assess correlations between perceived organizational justice and federal employees' work-related attitudes including job satisfaction, trust toward their supervisors and management, and intention to leave their agency [35]. The study found that higher levels of three types of organizational justice (distributive, procedural, and interpersonal) are positively related to job satisfaction and trust in supervisor and management, whereas they are negatively associated with turnover intentions of employees. Other work has found correlations between job satisfaction and perceptions of reciprocity between employees and employers [36]. These studies are suggestive of relationships between employees' workplace experiences and perceptions of integrity of organizational management.

Limited work has focused on organizational justice, ethics, and integrity in the federal workforce. The best predictors for maintaining scientific integrity in the workplace are not definitively known. However, based on prior relevant studies [17, 36–37], surveys [38–44], and federal scientific integrity policies [45], maintaining scientific integrity in the workplace depends on multiple factors that fall within the following three categories: 1) The existence and implementation of a clear comprehensive scientific integrity policy at the agency or department level; 2) the perception of competent and trustworthy agency leadership; and 3) an environment where employees feel effective and valued. A better understanding of what factors are most critical to maintain scientific integrity could provide government leaders and staff with information identifying the means by which political interference occurs within their agency. Such information may help to better inform the agency's scientific integrity policy, which could lead to more effective scientific work and policy decisions informed by science.

One way to assess the degree and dimensions of scientific integrity at federal agencies is through surveys of government scientists. The nonprofit Union of Concerned Scientists, working with Iowa State University's Center for Survey Statistics & Methodology, has conducted nine such surveys every 2–3 years since 2005 to assess the status of policies and practices at agencies relating to scientific integrity and to identify problem areas that might not otherwise be made public [38]. In the most recent study, conducted in 2018, we surveyed federal agency scientists' perceptions of scientific integrity under the Donald J. Trump administration, which is largely viewed by the scientific community and scholars as an administration with an unprecedented frequency of scientific integrity violations relative to prior administrations [17, 37, 46]. Little research has focused on perceptions of government scientists on scientific integrity over time and across agencies, particularly under the Trump administration. Therefore, surveying under the Trump administration offers a rare opportunity to gauge scientists' perceptions of how scientific integrity is best maintained at their agency. Moreover, our study provides insights into how these perceptions compare across multiple agencies and administrations using prior surveys with similar items addressing scientist's perceptions of scientific integrity. Using survey data collected under multiple administrations, we addressed the following questions: 1) What scientific integrity factors (e.g., adherence to scientific integrity policy, perception of effective leadership, censorship) are perceived to be most at threat in 2018 (i.e., under the Trump administration)? 2) Do perceptions of scientific integrity factors vary across federal agencies? 3) Have perceptions of scientific integrity changed over time?

## Materials and methods

### The survey instrument

The 2018 survey instrument consisted of 52 core questions, most of which were used on previous surveys, with a few revisions made to accommodate the wide range of agencies being surveyed (See supplementary material). Initial screening questions verified the respondent's primary agency and sub-agency (office) affiliation as well as the percentage of job duties involving science. If respondents indicated their job duties did not involve science, they were not permitted to complete the remaining questions. In addition to a core set of items answered by all agencies, the survey also included agency-specific questions that were answered based on the primary agency selected in the initial screening. With the exception of the screening items, survey questions were considered response optional and, in most cases, provided a "prefer not to disclose" answer option.

In this analysis, we identified and analyzed a set of ten questions that fit within three categories likely to be predictive of the state of scientific integrity within federal agencies (see Table 1

**Table 1. Factors contributing to scientific integrity perceptions and relevant survey questions.**

| Factor Contributing to Scientific Integrity Perceptions | Relevant Survey Questions |
|---|---|
| **Competent and trustworthy leadership** | 1. In your opinion, what are the greatest barriers to science-based decisions at your agency? (Choose up to 3.) Absence of leadership with needed scientific expertise Influence of political appointees in your agency or department* |
| | 2. The level of consideration of political interests hinders the ability of my agency to make science-based decisions. (Agree/Disagree scale) |
| | 3. The level of consideration of business interests hinders the ability of my agency to make science-based decisions. (Agree/Disagree scale) |
| **Feeling effective and valued** | 4. Over the past year, I have noticed that resource allocations (e.g., funding, staff time) have been distributed away from programs and offices whose work is viewed as politically contentious. (Agree/Disagree scale) |
| | 5. In the past year, I have been excluded from discussions or decisions related to my scientific work that I normally would expect to be a part of. (Agree/Disagree scale) |
| | 6. I have avoided working on climate change or using the phrase "climate change," though I was not explicitly told to avoid them. (Agree/Disagree scale) |
| | 7. Compared to one year ago, the effectiveness of your division or office has: (Decreased/Stayed the Same/Increased) |
| | 8. The presence of top agency decision makers who come from regulated industry inappropriately influences the decisions made by the agency. (Agree/Disagree scale) |
| | 9. Currently, I can openly express any concerns about the mission-driven work of my agency without fear of retaliation. (Agree/Disagree scale) |
| **Clear, comprehensive scientific integrity policy** | 10. My agency adheres to its scientific integrity policy. (Agree/Disagree scale) |

* On question 1 survey respondents were given 14 options; however, for the purposes of assessing perceptions of scientific integrity, only the two options listed below were included in this analysis.

for questions and categories). Other questions from this survey instrument and frequency data are provided in supplementary materials.

This table highlights the ten survey items (out of 52 core items total) discussed in this paper. These ten items were chosen as they are representative of three categorical factors evidenced to affect perceptions of scientific integrity in the workplace. The items are numbered by the order they are discussed in this paper, and do not correspond to their number or the order that they were addressed in the original administered survey instrument.

## The survey sample

Sixteen federal agencies within six departments were identified for the 2018 survey sample based on their science-based missions, commitment to scientific integrity, or history of past scientific integrity challenges. These agencies included the US Environmental Protection Agency (EPA); the National Oceanic and Atmospheric Administration (NOAA) and the US Census Bureau within the Department of Commerce; the Centers for Disease Control and Prevention (CDC) and the Food and Drug Administration (FDA) within the Department of Health and Human Services; the Energy Efficiency and Renewable Energy office (EERE) at the Department of Energy (DOE); the Bureau of Ocean Energy Management (BOEM), Bureau of Safety and Environmental Enforcement (BSEE), the National Park Service (NPS), US Fish and Wildlife Service (FWS), and the US Geological Survey (USGS) in the Department of the Interior (DOI); the Agricultural Research Services (ARS), the Economic Research Service (ERS),

the National Institute of Food and Agriculture (NIFA) and the National Agricultural Statistics Service (NASS) in the US Department of Agriculture (USDA); and the National Highway Traffic Safety Administration (NHTSA) at the Department of Transportation.

Staff email addresses for these agencies were obtained through publicly available online staff directories and Freedom of Information Act (FOIA) requests. Online directories for the FDA, CDC, USGS, NIFA, and ARS included both job titles as well as contact information.

For government agencies with an incomplete online employee directory or no directory at all, FOIA requests were filed. The offices where a FOIA request resulted in full staff lists were BOEM, BSEE, NPS, FWS and NHTSA.

For agencies with job title information available, employees were identified as scientific or nonscientific based on job title and office within their department. For the purposes of this survey, scientists were considered anyone whose job involved a significant level of science, including but not limited to research, analysis, modeling, inspection and oversight, and science policy. Full-time federal employees, contractors, and associates were included in the survey but fellows, students, and interns were excluded. When available, the specific office the employee worked in was used to exclude large amounts of people that were unlikely to perform the above scientific functions. Common non-scientific offices such as administration, finance, information technology, and facility maintenance were consistently excluded from lists.

The EPA, DOE, and Census Bureau were unresponsive to FOIA requests within six months. For these agencies, lists were obtained either from inside or external sources. A website called FederalPay.org was used to collect information on federal employees earning salaries exceeding $100k, allowing some contact information for NASS employees to be obtained. In early 2017, the EPA and DOE removed their employee directory from their website. However, the Data Refuge Project preserved the full EPA staff list in a publicly accessible database scraped in February 2017, though without job title information. Ultimately no job title screening information was available for the EPA, DOE, and the Census Bureau; thus, email addresses for all employees in relevant divisions were included in the sample. As a result, it is acknowledged that the sample likely included many ineligible individuals from these agencies. The final sample for the 2018 survey consisted of 63,248 email addresses. Survey contacts were made exclusively by email.

As an additional quality control mechanism, the survey instrument asked respondents to indicate what percentage of their time was spent on science. Respondents answering zero percent were routed to the final survey question and excluded from aggregate survey statistics.

## Survey administration

Prior to launching the survey, researchers contacted each of the sampled federal agencies, notifying them that the survey was being conducted. CSSM staff obtained survey approval from the Iowa State University Institutional Review Board (IRB #18–017).

The survey was prepared and made available to participants in one of three response modes: online Web survey, on paper using a downloadable PDF, or by telephone. Unique access codes were assigned to each person within the sample to ensure the integrity of the survey and its results.

The online survey was programmed using Qualtrics software. Respondents could access the survey login page either by clicking on the embedded link in their email invitation or by manually entering the URL. Respondents then entered their assigned access code to access their personal survey. As an additional layer of security, CSSM used Qualtrics' "Anonymizing Responses" feature, which removes all identifying respondent information (including email address, IP address, location data, and password/external data reference codes) upon

completion, resulting in data that cannot be linked to a particular respondent or email address. Both the survey and the data were stored on secure Iowa State University servers.

A PDF version of the survey was created by CSSM, which mirrored the online survey as much as possible. Agency-specific information embedded in the online survey was changed to be more generic for the PDF version, and routing or skip instructions were added. The survey email invitations and reminders provided links to both the CSSM and UCS Federal Scientists Survey webpages, which housed the PDF file for downloading. Recipients who wished to complete the PDF version were invited to print a copy and complete it on paper, returning it either by mail or by emailing a scanned file. For survey integrity purposes and to avoid duplicate responses, PDF respondents were asked to write their assigned access code on their completed survey. Once received, CSSM staff verified that the access code hadn't previously been used and then recorded the data. Once the PDF responses were entered, paper copies were securely destroyed and scanned forms were deleted.

Confidential phone interviews were also made available to survey recipients. A toll-free phone number was provided in the email invitations/reminders and on the CSSM and UCS survey webpages. Respondents were invited to call the number and schedule a time for the interview. Interviews were conducted by CSSM's professional survey staff. For verification and survey integrity purposes, phone respondents were asked to provide their assigned access code before completing the interview.

The email invitation and four email reminders each included an embedded survey link and the respondent's access code, as well as instructions for each of the three survey modes (See supplementary material). In addition, each email described the survey's purpose, the organizations involved, participant selection criteria, and anticipated time requirement of 15–20 minutes. The emails stated that sampled agencies had been notified about the survey and encouraged participants to complete the survey on their own personal time and equipment. An "opt-out" link was included for those who preferred not to participate or be contacted again. CSSM and UCS website links were provided for those who desired more information.

The CSSM and UCS websites both contained additional resources, including a frequently asked questions document, a link to the survey PDF, instructions for completing the survey by phone, and information about past research conducted by the organizations.

In the field of survey research, it is generally acknowledged that offering multiple response methods simultaneously does not necessarily increase response rates and can even lead to a decline in some situations. In this situation, however, given the subject matter and the anticipated concerns regarding confidentiality and anonymity, it was decided to allow respondents to choose the response option that best fit their comfort level. Multiple contact methods were not possible since only email addresses were available. Due to response collection limitations and other facters raised in the Discussion section, results should not be considered representative, but rather a window into perspectives of a subset of scientists at each agency.

## Data collection

The data collection period was February 12, 2018 –March 26, 2018. Email invitations were sent to a quarter of the sample per day on four successive days, with reminder emails sent approximately 6 to 9 days apart. Throughout data collection, respondents could enter their online survey using their access code as often as they wished until the survey was completed. Completed surveys were closed and immediately stripped of all personal identifiers, including email address, IP address and access code. During data collection CSSM and UCS staff received and responded to questions, comments, or concerns about the survey. Issues included

verification of survey legitimacy, questions about procedures, ineligibility (e.g., no scientific work), and technical problems.

Survey response varied by agency. Of the 63,248 email invitations sent, 1843 were undeliverable, 122 were classified as ineligible, and 1951 refused or selected the "opt-out" option. Data was obtained from 4211 respondents, including 3783 complete and 428 partially complete surveys. Of the 4211 surveys in the data set, 4178 were completed online, 23 on paper, and 10 by phone. The overall response rate was 6.67%, based on 4211 surveys out of 63,126 potentially eligible contacts. Response rates by agency ranged from 2.17% to 19.12% (see supplementary materials).

## Statistical analysis procedures

Statistical analyses were conducted across agencies for the 2018 survey and over time within agencies by referencing past survey data. All procedures were accomplished by using SAS and R.

The assessments across agencies were restricted to eight survey items for five agencies with the higher survey response rates: CDC, FDA, EPA, NOAA, and DOI (USGS, FWS, and NPS combined), hereafter referred to as the "five agencies analyzed." One of these ten items of interest uses nominal responses (Yes/No). The others have ordinal scale responses, excluding "Prefer not to disclose," "Don't know," and "Not Available" options.

The nominal response item (Question 1 in Table 1) has 16 sub-questions, or options, to be selected (yes) or not selected (no). For each sub-question, the association between agencies and the response was assessed by using the Pearson Chi-square test. An overall p-value for each sub-question was also calculated by simulating data according to the observed probability within each agency and its fixed sample size. A post-hoc test was conducted after significant overall test and Tukey adjustment [47] was enhanced to control family-wise type I error (FWE).

For questions with ordinal responses, Pearson Chi-square and Mantel-Haenszel Chi-square tests were utilized to assess the overall association and linear trend. A Cochran-Armitage Trend test was applied for post-hoc pair-wise comparisons and a Tukey adjustment was used to control FWE.

The analyses of responses over time were restricted to the agencies: CDC, EPA, FWS, FDA, and NOAA for several comparable question items. Most agencies only had survey data available from 2015 and 2018, except for the FDA which had data from earlier surveys for some items. Pearson Chi-square tests were used for evaluating overall association between time and responses, and Mantel-Haenszel Chi-square tests were applied for ordinal responses. Since the FDA had available data from more than two surveys, post-hoc comparisons were conducted when needed and Hochberg (step-up Bonferroni) was applied for FWE control.

## Results and discussion

We examined the degree to which over 3,700 federal scientists working across five federal agencies perceive scientific integrity to be upheld at their agencies by analyzing responses to survey questions that inform the three categories identified as contributing to maintaining scientific integrity in the workplace (see Table 1 for questions and categories analyzed). We also compared responses over time using survey data collected under prior administrations for the CDC, FDA, EPA, NOAA, and the FWS. Overall, we show that (a) federal scientists in some agencies perceived the influence of political officials and the absence of leadership with needed scientific expertise as major barriers to fulfilling their agencies' science-based mission, (b) federal scientists vary in the degree to which they feel valued and effective in their work varied

across agencies, and (c) in comparison to prior survey data, federal scientists perceived that the effectiveness of their offices/divisions decreased under the Trump administration. However, federal scientists from some agencies also perceived that their agency adheres to its scientific integrity policies under the Trump administration.

## Competent and trustworthy leadership

Federal scientists from the five agencies analyzed largely perceived that their current agency leadership lacks technical expertise and is not perceived as trustworthy with respect to scientific work. Across the five agencies, 2026 survey respondents reported "influence of political appointees" and "absence of leadership with needed scientific expertise" as major barriers to science-based decisions made at their agency (Fig 1). This is an important finding given current research that shows leadership's technical competence and employees' trust in leadership can affect employee's job satisfaction, engagement in work, and productivity [48, 49]. For example, in a study that surveyed 35,000 US and UK employees, technical competence of leaders was a strong predictor of job satisfaction [49]. In another study of 204 employees working in various organizations in South Africa, worker perception of leaders as trustworthy and who maintain integrity in their work was positively associated with worker engagement and productivity [50]. If leadership cannot fully relate to the tasks their employees undertake and are not viewed as trustworthy to their employees and the work they perform (e.g., politically interfering in scientific work), we would expect a high number of employees to view the absence of these types of leaders as a barrier to their work. We also would expect lower job satisfaction

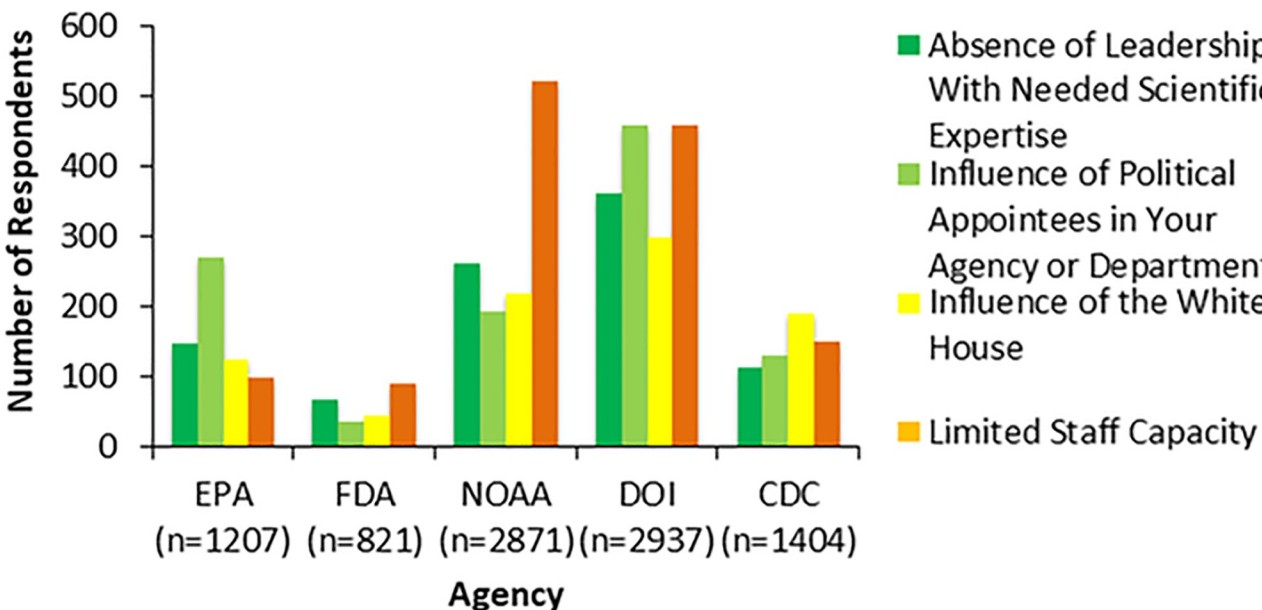

**Fig 1. Barriers to science-based decisions at agencies.** Figure shows the results from survey respondents' selections to the survey item, "In your opinion, what are the greatest barriers to science-based decisions in your agency?" Survey respondents could choose up to three barriers out of 14 options: delay in leadership making a decision; absence of leadership with needed scientific expertise; uncertainty or disagreement with the science; influence of political appointees in your agency or department; influence of the White House; influence of Congress; influence of other agencies; influence of industry stakeholders; influence of nongovernmental interests (such as advocacy groups); inefficient decision making process within the agency; potential discrepancy with existing rules and regulations; uncertainty of agency jurisdiction; complexity of the issue; limited staff capacity; other, and prefer not to disclose. The top five barriers selected were: absence of leadership with needed scientific expertise, influence of political appointees in your agency or department, influence of the White House, limited staff capacity, and delay in leadership making a decision.

and effectiveness as a byproduct if such leadership were perceived to be lacking (discussed in below section on perception of being valued).

Among the five agencies analyzed in this study, we also found qualitative evidence from open-ended response questions that some federal scientists perceive their agency leadership to be technically incompetent and untrustworthy to their work. Respondents were asked, "How have actions taken or changes made by the current administration related to science (positive or negative) helped or harmed your agency's mission?" In response, an anonymous employee from DOI stated, "Appointing a secretary with little understanding or appreciation of our agency's function or structure has been detrimental to work productivity." When prompted with the same question, an anonymous EPA employee replied, "Top levels of leadership don't fundamentally believe in the value and mission of the agency and are actively planning to hit the agency, while lying about their intentions. Top leadership appoints political hacks who utterly lack scientific credentials, and who don't understand or value the scientific process."

While most respondents across the five agencies analyzed here chose "influence of political appointees" and "absence of leadership with needed scientific expertise" as a major barrier to science-based agency decisions, there was variation across agencies in what federal scientists considered the greatest barriers to their work (Fig 1). A greater proportion of respondents from the EPA than other agencies cited "influence of political appointees" as a top barrier to science-based decisions, with 60% of EPA respondents choosing this option. At DOI, this was also the most frequently chosen barrier, selected by 40% of respondents. At the FDA and NOAA this option was chosen in lesser proportions, with 10% of FDA respondents and 17% of NOAA respondents. Both the chi-square p-value and simulated p-value were <0.0001 and all five agencies responded differently except CDC vs. NOAA (21% vs. 17%, Tukey adjusted p-value = 0.097). The option "absence of leadership with needed scientific expertise" was chosen at higher proportions at EPA (33%, 147 respondents) and DOI agencies (31%, 361 respondents) compared to the FDA (18%, 65 respondents) and NOAA (23%, 260 respondents).

Other potential barriers selected most often by respondents included: (1) "influence of the White House." The average selection rate for all five agencies was 23%. The FDA (11.9%) and NOAA (19%) chose this option at a significantly lower rate than other agencies (31% for CDC, 27% for EPA, and 26% for DOI) with p-values <0.02. (2) "Limited staff capacity." The average rate among the five agencies for this option was 35%. The rate at which this option was chosen for the CDC, EPA, FDA, NOAA and DOI were 25%, 22%, 25%, 45%, and 40%, respectively. NOAA had a significantly higher selection rate for this option compared to other agencies, except versus the FDA (p-value = 0.27). The EPA had a significantly lower rate for this choice compared to the DOI (p-value = 0.04). (3) "Delay in leadership making a decision." The average selection rate among all five agencies for this chosen option was 22% with no significant differences in rate between agencies (p-value = 0.25).

Other survey items also provide evidence that some federal scientists do not currently perceive agency leadership as trustworthy. Many respondents did not feel that they could trust agency leadership to keep political interests from influencing their works' role in decisions. For example, 83% of respondents (345 respondents) at the EPA and 58% of respondents (621 respondents) at the DOI agreed or strongly agreed that consideration of political interests hindered their agency's ability to make science-based decisions, while 48% of respondents (255 respondents) at the CDC, 32% of respondents (101 respondents) at the FDA, and 42% of respondents (462 respondents) at NOAA agreed with this statement (Fig 2). Similar results were found with respect to levels of consideration of business interests with 75% of respondents (317 respondents) at EPA, 46% of respondents (457 respondents) at DOI agencies, 36% of respondents (114 respondents) at the FDA, and 38% of respondents (416 respondents (38%) at NOAA indicating that these interests serve as a barrier to science-based work.

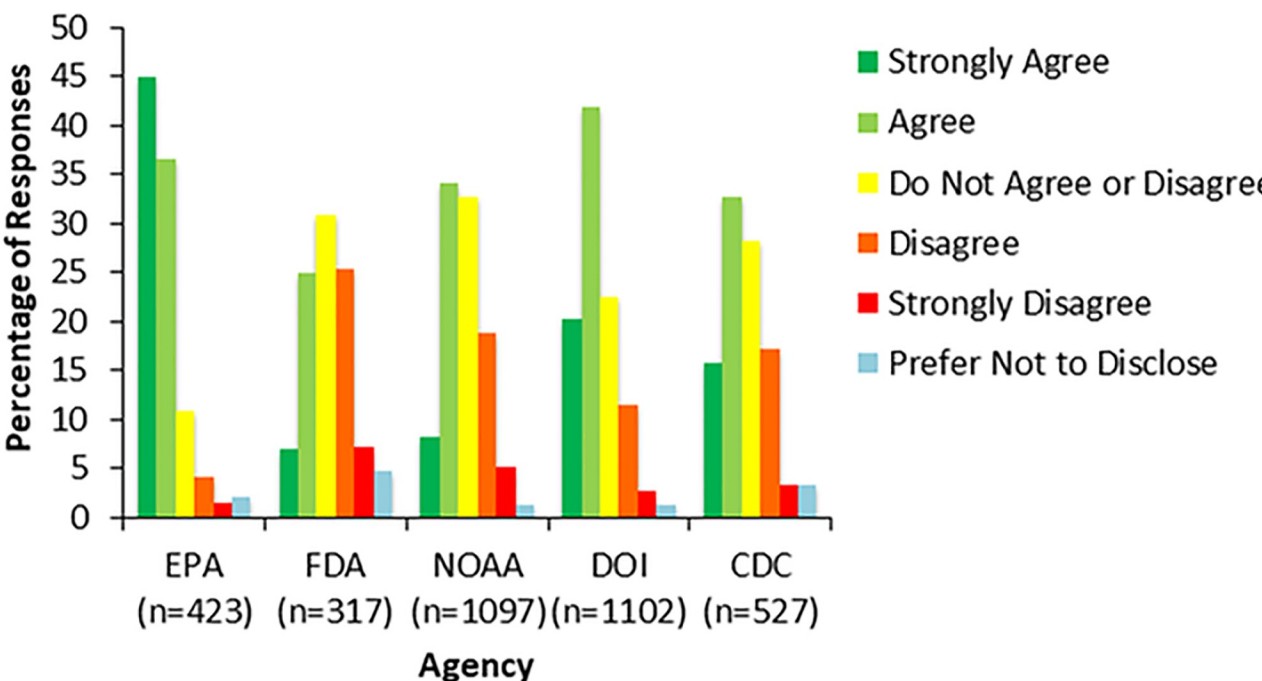

**Fig 2. Consideration of political interests in science-based decisions.** Figure shows the results from survey respondents' responses to the survey item, "The level of consideration of political interests hinders the ability of my agency to make science-based decisions." Survey respondents could choose one of six options: strongly disagree, disagree, do not agree or disagree, agree, strongly agree, or prefer not to disclose. Responses varied across agencies with more scientists agreeing that political interests hinder their agency's science-based decisions at the EPA and DOI as compared to NOAA and the FDA.

The items discussed above provide a measure of scientific expert's perception of their scientific work being politicized. The responses to these items indicate that survey respondents at the EPA and DOI do not perceive agency leadership to be trusting of scientific work when compared to political interests, which scientists perceive to be dominating decisions. Additionally, survey respondents perceive leadership to lack the technical expertise to be involved in science-policy work. Prior work has shown that lack of technically competent and trustworthy leadership can strongly affect employee's job satisfaction and engagement and these two factors often affect employee work productivity. While a majority of respondents at the EPA and DOI reported that technically incompetent leadership as well as political interests were major barriers to their work, there was variation in responses across the five agencies analyzed. At the CDC, the FDA, and NOAA scientists appear to be more trusting of agency leadership to defend their work against political interests and to be more technically competent.

The perception of leadership at the CDC, FDA, and NOAA having technical competence and behaving in a way that scientists could trust their leaders to defend their work was evidenced in open-ended responses. Qualitative data was in-line with survey frequency data regarding federal scientists' perceptions of leadership and how those perceptions varied across agencies. For example, when prompted with the question, "Is there anything else you would like to share with us regarding scientific integrity at your agency?" an anonymous FDA employee stated, "I believe that at FDA we have been exceedingly fortunate with Dr. Scott Gottlieb. He is an ethical scientist steeped in policy formulation. He has been tasked with continuing a tradition of upholding scientific principles and procedures and I believe he is doing an outstanding job despite the current Congressional climate." Responding to the same prompt,

another anonymous FDA employee said, "I've been pleasantly surprised by Commissioner Gottlieb's knowledge and focus on FDA science. I was at a brief with him and he was interested in the science and less focused on the legal and political effects than I would have guessed." The technical competence of FDA Commissioner Gottlieb was signaled as positive across many open-ended responses in our survey, and many scientists expressed they trusted the commissioner to defend and use their scientific work. Similar open-ended responses were provided by NOAA employees, for example, "I have the sense that our NOAA leaders are working so hard to keep our agency focused on science and mission." At the time that the survey was administered, CDC lacked a politically appointed leader, but scientists generally reported positively on how career leadership handled the science policy process.

Qualitative data was in-line with survey frequency data regarding federal scientific experts' perceptions of leadership and how those perceptions varied across agencies. We would expect that such variation would also be reflected in workers' perceptions of their value and effectiveness.

## Perception of value in the workplace

While results varied across agencies, we find that scientists reported feeling less valued and effective in the workplace under the Trump administration.

There is increasing evidence that employees' perceptions of their value in the workplace strongly affect their engagement [51]. While there is no single definition of employee engagement, the term can be generally defined as the emotional attachment of employees to their place of work [51–53]. If employees are more engaged in their work, it is likely that this will lead to a more effective and productive workforce [54].

Prior research has identified several workplace attributes that contribute significantly to employee engagement. One attribute common across many studies is feeling valued and involved. In a survey that asked 1,714 employees about the perception of their value and engagement in their work, of the 93 percent of employees reporting they felt valued by their employer, 88 percent of those employees also reported being engaged in their work [55]. Another identified driver of employee engagement is being kept abreast of what is going on in the organization, especially regarding communication of issues related to their specific work [56]. Employee finances are also relevant to the perception of value in the workplace. While this includes the personal financial aspect of an organization's employees (e.g., adequate pay, performance-based raises), it also includes the allocation of adequate financial resources to an employee's work projects. When individuals receive resources from their organization to conduct their work, they feel obliged to respond in kind and "repay" the organization [57].

Based on the above research, we analyzed four items in our survey that relate to attributes of employee engagement and employee effectiveness. We included a measure of effectiveness in this category, since we would expect, based on prior studies, that effectiveness is strongly related to employee engagement. The four items we analyzed were: 1) the employee receiving resources to do their work, 2) feeling included in team discussions when the employee's work is involved, 3) the employee being able to speak freely about their scientific work even when it is viewed as politically contentious (e.g., 'climate change'), and 4) direct reporting of how the employee perceives the effectiveness of their office or division (questions 4, 5, 6, and 7, respectively, listed in Table 1).

Overall, across the four items analyzed, we find evidence that employee engagement of federal scientists varied across the surveyed agencies. On question 4, some respondents reported that over the past year they have noticed that resource allocations (e.g., funding, staff time) have been distributed away from programs and offices whose work is viewed as politically

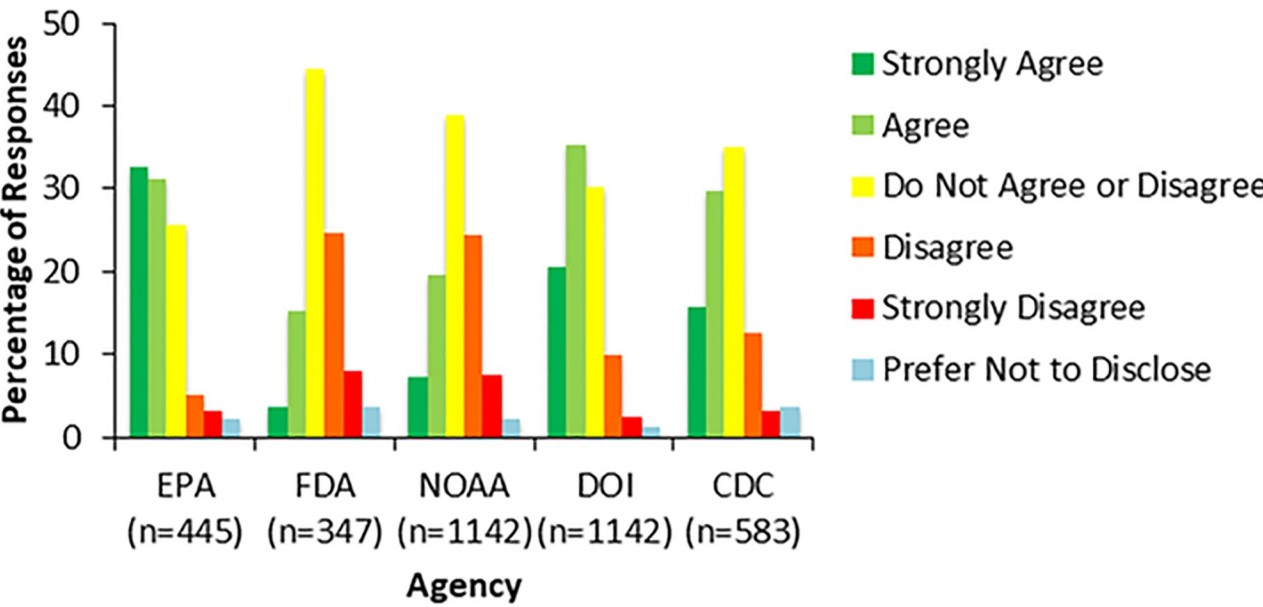

**Fig 3. Resource allocations for politically contentious work.** Figure shows the results from survey respondents' responses to the survey item, "Over the past year, I have noticed that resource allocations (e.g., funding, staff time) have been distributed away from programs and offices whose work is viewed as politically contentious." Survey respondents could choose one of six options: strongly disagree, disagree, do not agree or disagree, agree, strongly agree, or prefer not to disclose. Responses varied by agency with a larger percentage of scientists agreeing that resource allocations have been distributed away from programs viewed as politically contentious at the EPA and DOI as compared to NOAA and the FDA.

contentious. A higher percentage of respondents from the CDC, DOI and EPA perceived a lack of resources for their work, with 47% (264 CDC respondents), 57% (646 DOI respondents) and 65% (284 EPA respondents), agreeing or strongly agreeing that this was the case (Fig 3). In comparison, 20% of FDA respondents and 28% of NOAA respondents either agreed or strongly agreed that allocations have been distributed away from these programs. These responses were significantly different with p-values <0.0001 (DOI vs. FDA, DOI vs. NOAA, EPA vs. FDA, and EPA vs. NOAA, CDC vs. FDA, CDC vs. EPA, CDC vs. NOAA). Agreement was significantly higher for EPA respondents vs. respondents at DOI (p-value = 0.007) and CDC respondents vs. DOI respondents (p-value = 0.005).

On being provided adequate financial resources to do their jobs, many respondents, in their qualitative responses, cited real or anticipated budget cuts as inhibiting their job effectiveness. At the CDC, for example, several respondents cited potential budget cuts and resource constraints around infectious diseases and international cooperation as limiting their ability to do their jobs. One CDC respondent stated, "Talk of defunding global health initiatives has caused my division to start terminating research collaborations with international laboratories that function as infectious disease surveillance sites in Africa and Asia." Another CDC respondent echoed similar budget concerns, "Proposed funding cuts limits our [agency's] capacity for responding to infectious disease overseas and domestically. These cuts don't just affect our ability to prepare and respond at a federal level either. These will disproportionately affect smaller state and local health departments and grant funded programs."

Some respondents also reported being excluded from conversations they would normally be a part of (Table 1, question 5); however, this was not the case for most survey respondents. The percentage of respondents who disagreed with this statement varied by agency, with NOAA (61%, 671 respondents), FDA (61%, 193 respondents), and CDC (58%, 311

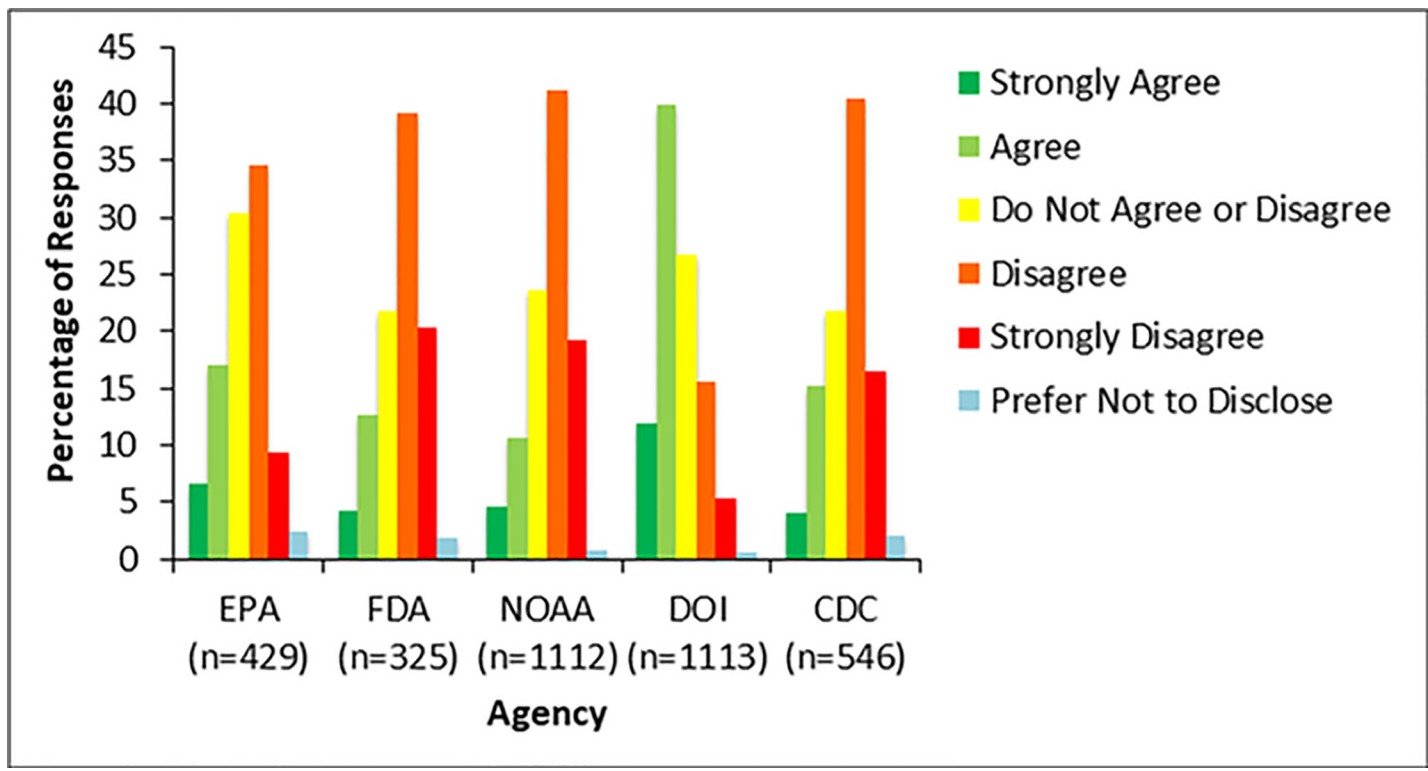

**Fig 4. Exclusion of scientists from discussions and decisions.** Figure shows the results from survey respondents' responses to the survey item, "In the past year, I have been excluded from discussions or decisions related to my scientific work that I normally would expect to be a part of." Survey respondents could choose one of six options: strongly disagree, disagree, do not agree or disagree, agree, strongly agree, or prefer not to disclose. Most respondents across the FDA, DOI, and NOAA disagreed with this statement. Respondents from EPA showed more agreement that they have been excluded from conversations that are related to their scientific work.

respondents) demonstrating the greatest disagreement among the agencies analyzed in this study (Fig 4). Respondents at the DOI (54%, 593 respondents) and EPA (45%, 188 respondents) disagreed or strongly disagreed to a slightly lesser extent. Responses at the EPA were significantly different from other agencies (p-values ≤0.03), and responses from NOAA were significantly different from DOI as well (p-value = 0.0001) (Fig 4).

Some scientists also reported self-censorship with respect to climate change-related work (Question 6 in Table 1, Fig 5). As observed with the other items in this category, responses varied across agencies with a higher perception of self-censorship at the EPA and DOI. At the EPA, 31% of respondents (124 respondents) agreed or strongly agreed that they had avoided working on climate change or using the phrase "climate change," though they were not explicitly told to avoid them, however a similar proportion of those surveyed at the EPA (34%, 134 respondents) also disagreed or strongly disagreed with the same statement. At the DOI, 26% of respondents (279 respondents) agreed or strongly agreed, and 48% of respondents (512 respondents) disagreed or strongly disagreed with the same statement. The distribution of responses was significantly different between EPA and DOI (p-value = 0.002). Although the percentage of respondents in agreement was not as high for NOAA respondents as at the EPA or DOI, 16% of respondents at NOAA (179 respondents) were in agreement that they avoided working on climate change issues. While the percentage of respondents reporting self-censorship is low, it is unusual for hundreds of federal scientists to report censoring science-based information. Therefore, we expect this to be a signal of devaluing employees at the EPA and DOI.

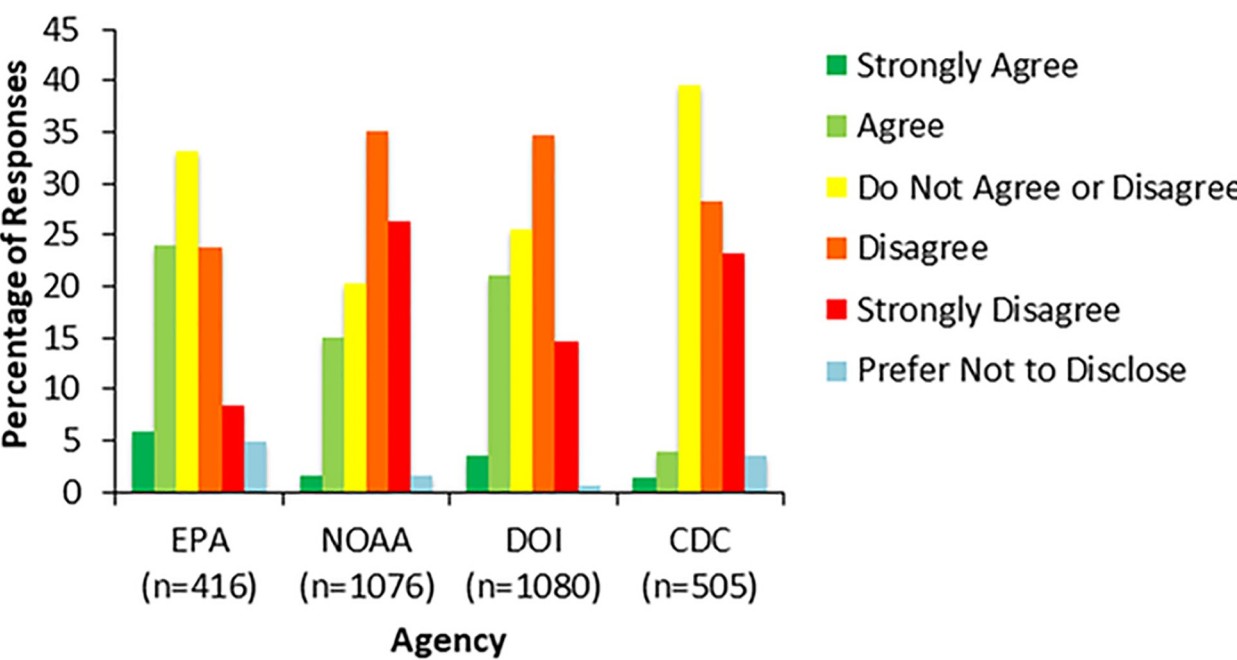

**Fig 5. Scientist self-censorship on climate change.** Figure shows the results from survey respondents' responses to the survey item, "I have avoided working on climate change or using the phrase "climate change," though I was not explicitly told to avoid them." Survey respondents could choose one of six options: strongly disagree, disagree, do not agree or disagree, agree, strongly agree, or prefer not to disclose. Respondents at the FDA were not asked to respond to this item as climate change is not a priority of their work. More survey respondents at the EPA and DOI agreed that they were self-censoring the phrase "climate change" in their work compared to NOAA respondents. The percentage of respondents reporting self-censorship was generally low across agencies; however, hundreds of scientists reported self-censoring a scientific term.

We did not see a similar pattern at the CDC with only 5% of respondents (25 respondents) agreeing that they had avoided working on climate change or using the phrase "climate change," though they were not explicitly told to avoid them. This response could be a product of employees feeling more valued at the CDC, or that climate change is not as broadly researched across the federal agency as compared to other agencies like the EPA or DOI.

Lastly, we analyzed the perception of office effectiveness by asking, "Compared to one year ago, the effectiveness of your division or office has: increased, stayed the same, or decreased." (Table 1, question 7; Fig 6) Respondents had the option to choose one response ("don't know," "not applicable," and "prefer not to disclose," answer options were also available). Perceived effectiveness and value are often intertwined as workers who are valued feel more engaged in their work and, therefore, are often more effective at their jobs. Therefore, it is possible that perceived effectiveness is an indicator of employees' perception of their own value and/or the value of other employees in the workplace. Perceived effectiveness significantly varied by agency with a majority of respondents reporting decreases at the EPA (68%, 284 respondents) and DOI (58%, 624 respondents), whereas the largest portion of respondents at NOAA (50%, 541 respondents), the CDC (43%, 228 respondents), and the FDA (54%, 170 respondents) reported that their office's effectiveness had largely not changed over the prior year (Fig 6). The CDC was different from FDA statistically, with a p-value of 0.001, even though the largest portion from both agencies reported no change in effectiveness. The EPA and DOI were also statistically different (p-value = 0.004) for all responses, even though they both reported a decrease in effectiveness.

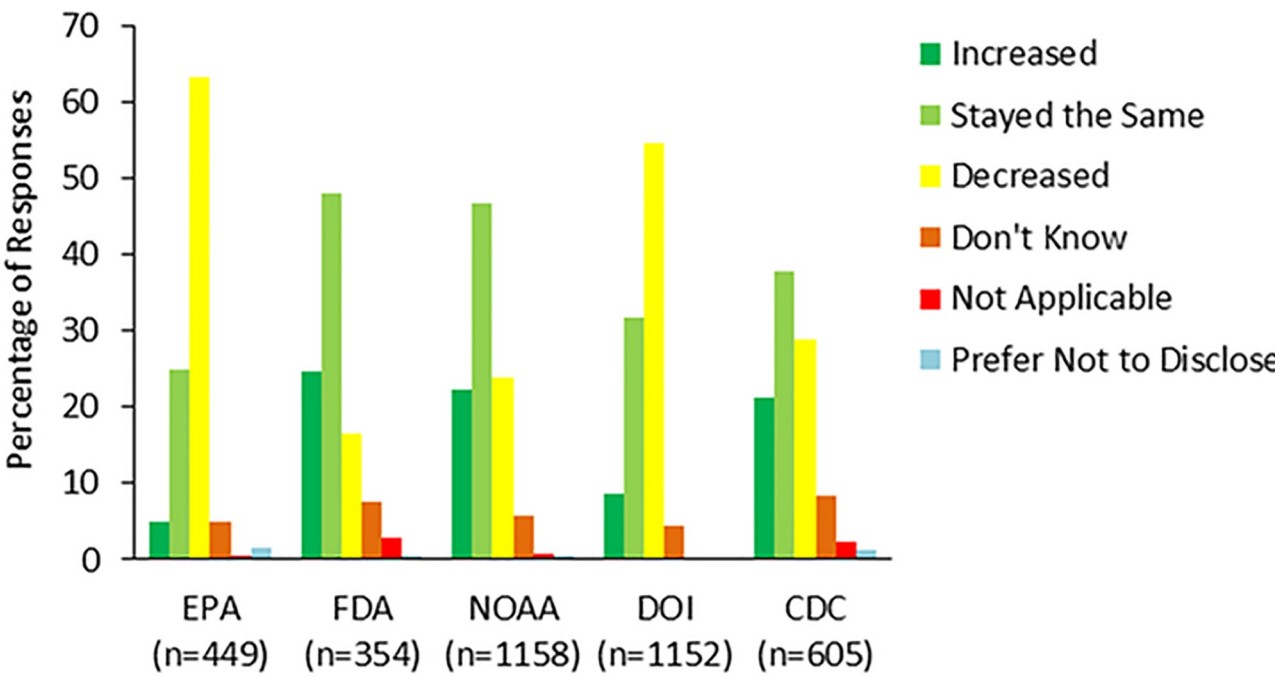

**Fig 6. Office effectiveness over time.** Figure shows the results from survey respondents' responses to the survey item, "Compared to one year ago, the effectiveness of your division or office has." Survey respondents could choose one of six options: increased, stayed the same, decreased, don't know, not applicable, or prefer not to disclose. Responses varied by agency with respondents at EPA and DOI largely reporting decreased job effectiveness, whereas most respondents at the FDA and NOAA reported their effectiveness had stayed the same.

These results are in-line with other items that show more responses from DOI and EPA respondents indicating that they may not feel valued by their leadership, while the opposite is true for CDC, FDA, and NOAA. Therefore, it would be expected that federal scientists at the EPA and DOI would perceive their agencies as not very effective compared to scientists at the CDC, FDA and NOAA.

## Comparisons over time

To provide insight into whether the perception of scientific integrity being upheld varied over time across political administrations, we compared 2018 results to prior survey studies for CDC, EPA, FWS, FDA, and NOAA. Ideally, we would make these comparisons for all survey items analyzed in this study (see Table 1). However, the language used in survey items has changed over time to ensure that we are asking respondents' questions that most effectively gauge their perception of scientific integrity. Additionally, many agencies surveyed in our current study were not included in prior surveys. Therefore, our comparative analyses are restricted to three questions that have the same or very similar language across surveys, and they also are restricted to agencies that have been surveyed at least once in a prior survey.

To determine how perceptions of leadership being competent and trustworthy have changed over time, we analyzed responses to the item, "The presence of top agency decision makers who come from regulated industry inappropriately influences the decisions made by the agency" (Table 1, question 8: Fig 7). We were able to compare responses across three agencies: The CDC (comparison across surveys implemented in 2015 and 2018), FWS (comparison across surveys implemented in 2015 and 2018), and FDA (comparison across surveys

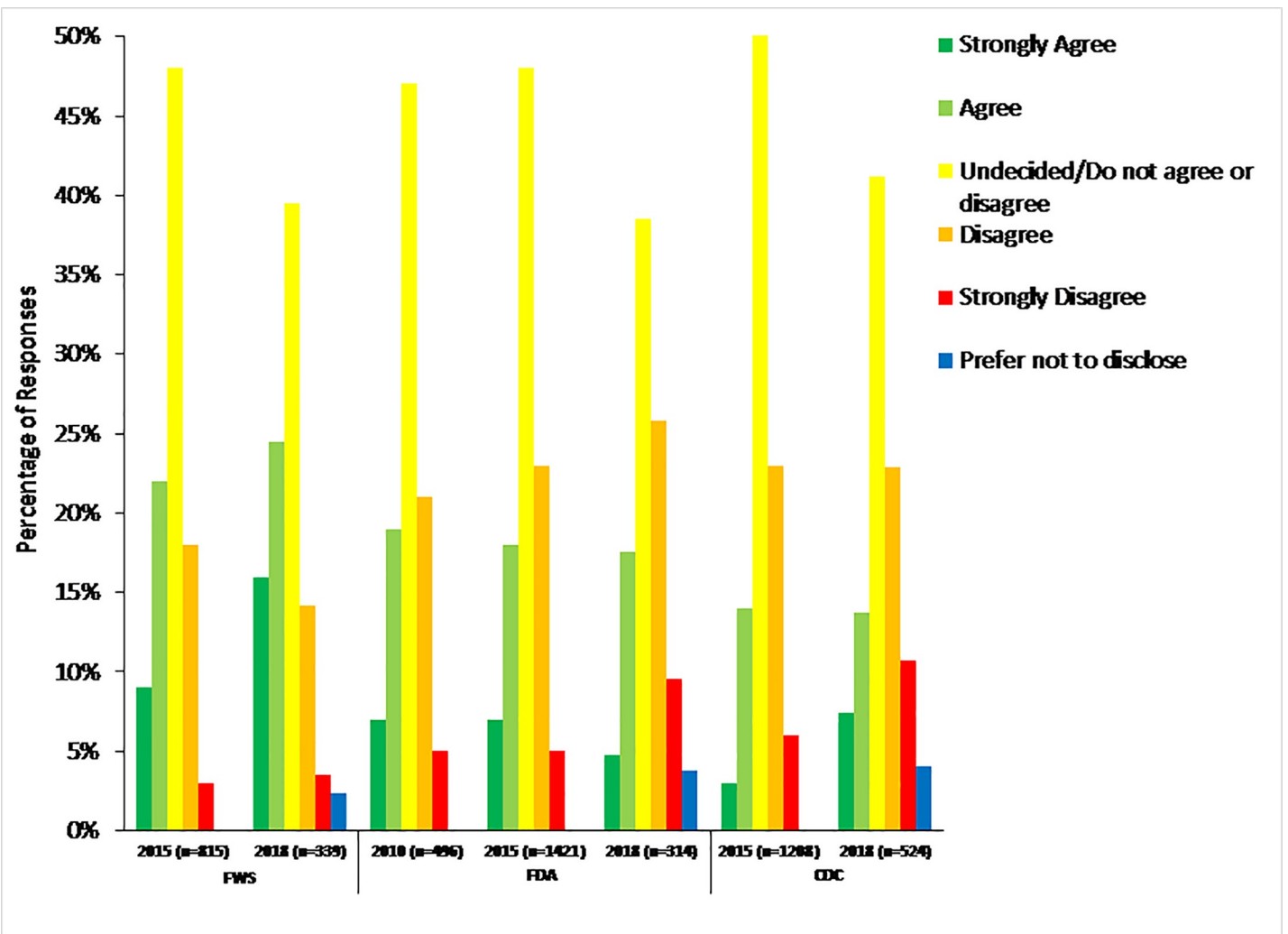

**Fig 7. Presence of decisionmakers from regulated industry.** Figure shows the results from survey respondents' responses to the survey item, "The presence of top agency decision makers who come from regulated industry inappropriately influences the decisions made by the agency." Survey respondents could choose one of seven options: strongly disagree, disagree, do not agree or disagree, agree, strongly agree, don't know, or prefer not to disclose. While responses significantly varied by agency, most responses skewed more positive over time according to Mantel-Haenszel Chi-square tests. Thus, more scientists perceive the presence of decision makers who come from regulated industries inappropriately influencing agency decisions.

implemented in 2010, 2015, and 2018). Responses varied over time for the CDC, FWS, and FDA with responses showing more agreement (agree or strongly agree) in the 2018 survey (i.e., indicating a greater level of inappropriate influence in 2018). At the FWS, the proportion of respondents who agreed with this item was significantly lower in 2015 (31%, 254 respondents) than in 2018 (40%, 137 respondents; Chi-square test p = 0.0012; Mantel-Haenszel Chi-square, p = 0.0012). We also found significant differences among responses at the FDA (p = 0.0064). A Mantel-Haenszel Chi-square test indicated more positive results in later surveys for this item at FDA (p = 0.0054), with agree/strongly agree responses slightly varying between 2010 (26%, 131 respondents), 2015 (25%, 346 respondents), and 2018 (23%, 70 respondents). Responses also varied significantly at the CDC (p<0.0001). Results skewed statistically more towards agreement (Mantel-Haenszel Chi-square, p<0.0001) in 2018 (21%, 111 respondents) than in 2015 (17%, 198 respondents).

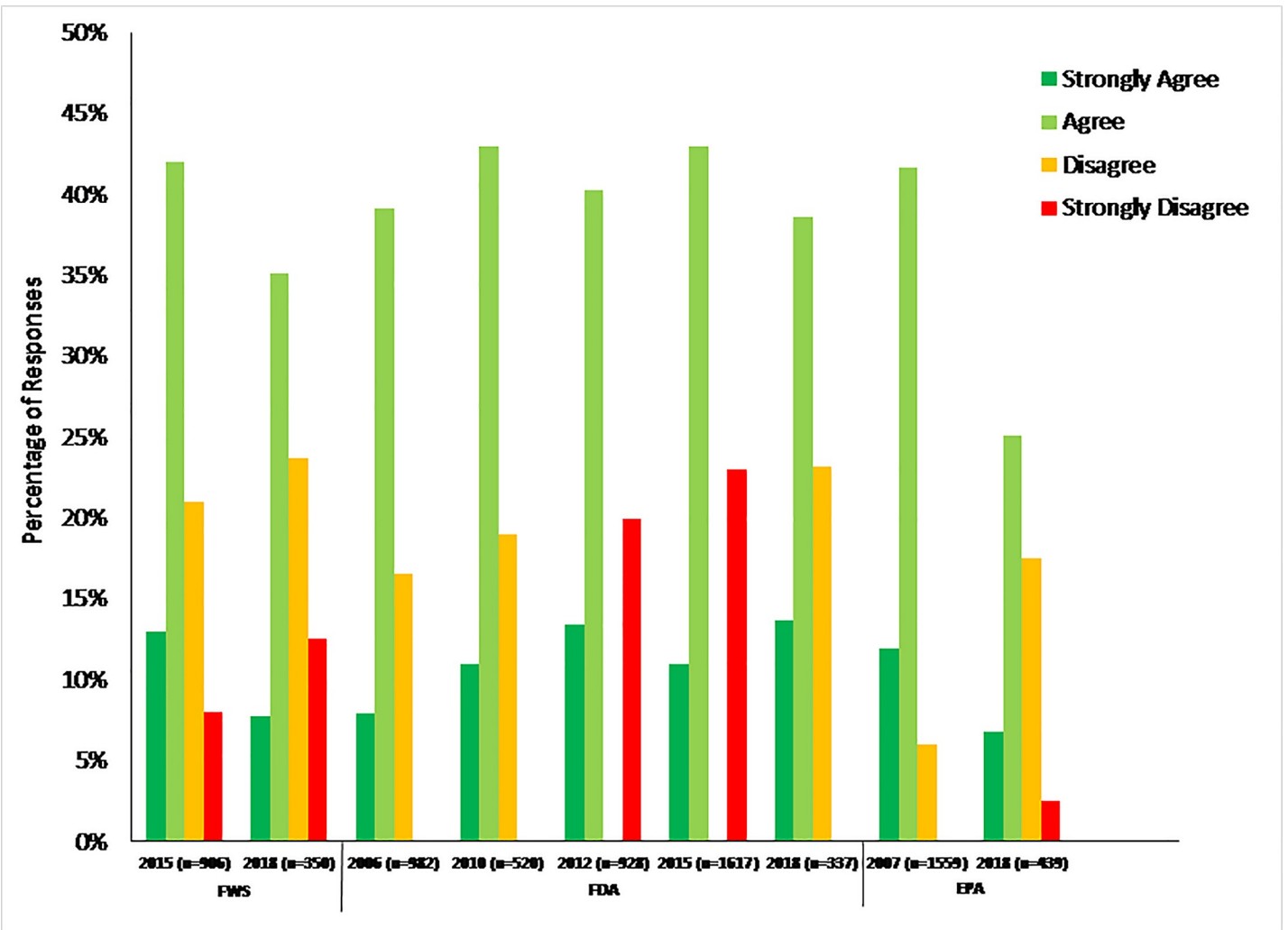

**Fig 8. Fear of retaliation.** Figure shows the results from survey respondents' responses to the survey item, "Currently, I can openly express any concerns about the mission-driven work of my agency without fear of retaliation." Survey respondents could choose one of seven options: strongly disagree, disagree, do not agree or disagree, agree, strongly agree, don't know, or prefer not to disclose. Mantel-Haenszel Chi-square tests indicated survey responses skewed more positive in later surveys both for the FWS and EPA but skewed more negative for later surveys at the FDA.

To assess if federal scientists perceived their value to have changed over time, we compared responses across surveys to the item, "Currently, I can openly express any concerns about the mission-driven work of my agency without fear of retaliation." (Table 1, question 9; Fig 8) Chi-square tests showed significant differences in responses for the EPA, FWS, and FDA; and Mantel-Haenszel chi-tests revealed that responses were more negative in later surveys as compared to earlier surveys (i.e., indicating a greater fear of retaliation in 2018). At the FWS, for example, 29% of respondents disagreed or strongly disagreed with this statement in 2015 whereas 36% of respondents disagreed or strongly disagreed with this statement in 2018 (Mantel-Haenszel Chi-square, p = 0.0011). At the EPA, disagree/strongly disagree responses also were significantly different in 2015 (24%, 382 respondents) versus 2018 (44%, 191 respondents; Mantel-Haenszel Chi-square, p<0.0001). While responses were significantly different among surveys implemented at the FDA (2006, 2010, 2012, 2015, and 2018), responses skewed more negative in earlier versus later surveys. In 2006, a higher percentage of respondents disagreed

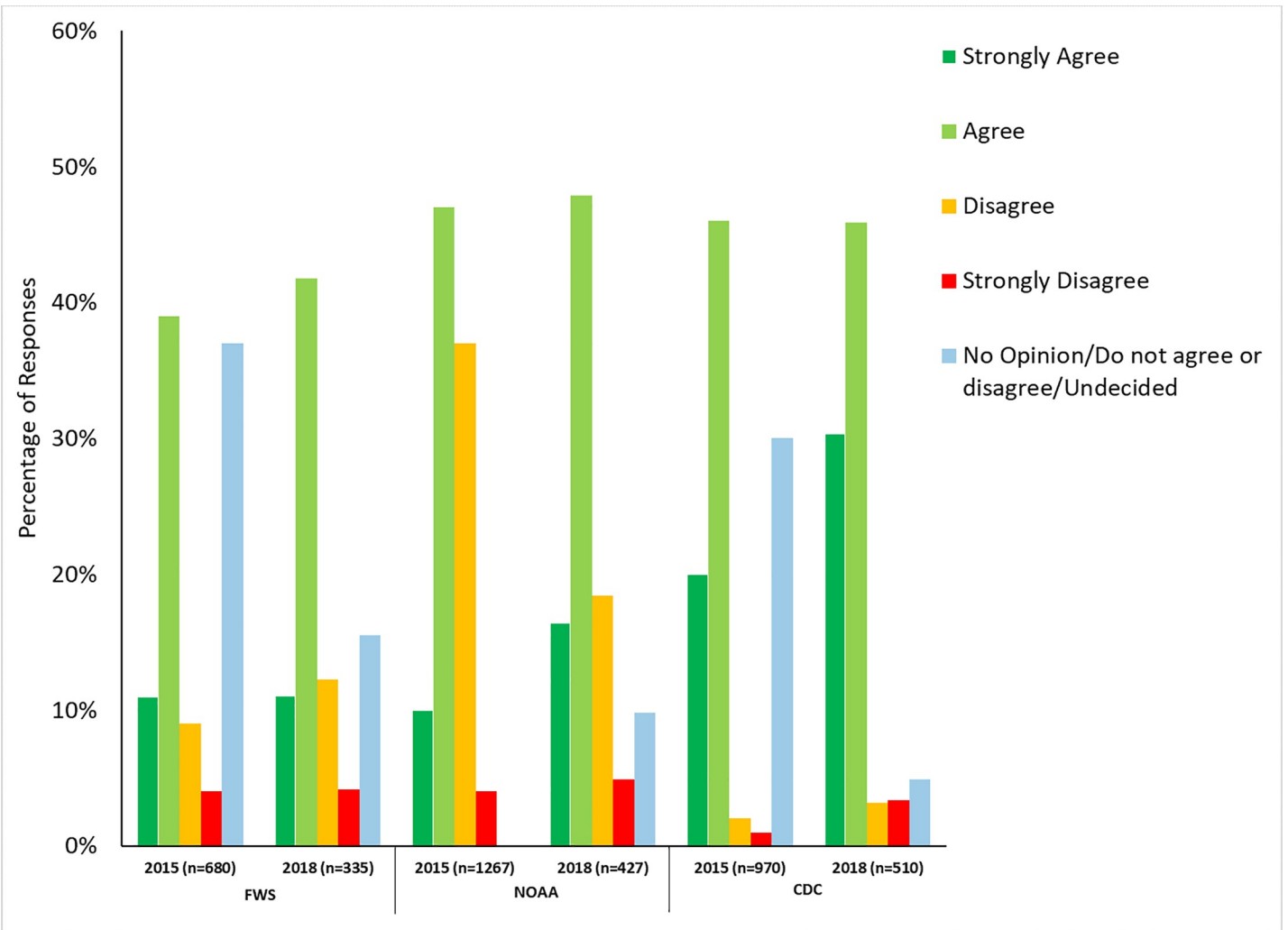

**Fig 9. Adherence to agency scientific integrity policy.** Figure shows the results from survey respondents' responses to the survey item, "My agency adheres to its scientific integrity policy." Survey respondents could choose one of seven options: strongly disagree, disagree, do not agree or disagree, agree, strongly agree, don't know, or prefer not to disclose. As compared to a survey conducted in 2015, agreement among scientists at the CDC and NOAA increased in 2018 regarding the agencies adherence to their scientific integrity policy, whereas responses trended towards a neutral/don't know response for the FWS.

or strongly disagreed (36%, 357 respondents) with this survey item as compared to all other years when the survey was implemented including 2018 (22%, 75 respondents; Mantel-Haenszel Chi-square, p<0.0001).

Only the CDC, FWS, and NOAA had questions from previous surveys that we could compare to determine how their agencies adhere to scientific integrity policies (Table 1, question 10; Fig 9). The CDC, FWS, and NOAA all responded positively over time to how their agencies uphold scientific integrity policies with those either agreeing or strongly agreeing changing from 66% to 76% in the CDC from 2015 to 2018 (p-value <0.0001), 50% to 63% in the FWS from 2015 to 2018 (p-value <0.0001), and 57% to 64% in NOAA (p-value <0.0001; Fig 9).

While it is not possible to compare responses for all survey items across time, the comparisons we are able to analyze provide further validation that our 2018 results, showing variation by agency, are sound. Across many agencies, we find that federal scientists perceive some aspects of scientific integrity (ethical leadership and feeling valued) to have decreased in 2018.

However, this is not the case at all agencies. For example, federal scientists at the FDA perceive scientific integrity to have been under greater assault during an earlier time. At other agencies, such as at the EPA, federal scientists are in significant agreement that some aspects of scientific integrity are at a loss in 2018.

## Adherence to agency scientific integrity policy

The 2018 survey asked respondents the extent to which they agreed or disagreed that "my agency adheres to its scientific integrity policy." (Table 1, question 10; Fig 9) Most scientists at the DOI, CDC, FDA, and NOAA reported that they agreed or strongly agreed that their agency adhered to its scientific integrity policy. Specifically, 76.1% at DOI (752 respondents), 87% at the CDC (389 respondents), 80% at the FDA (209 respondents), and 79% at NOAA (696 respondents) agreed or strongly agreed that their agency adheres to its scientific integrity policy. This compares to 12% (115 respondents), 7% (33 respondents), 7% (17 respondents), and 9% (78 respondents) at the DOI, CDC, FDA, and NOAA, respectively, disagreeing or strongly disagreeing that their agency adheres to its scientific integrity policy.

Among EPA respondents, however, 46% (144 respondents) agreed or strongly agreed that the agency adhered to its scientific integrity policy, while 31% (100 respondents) disagreed or strongly disagreed. This response from EPA scientists was significantly different from the other four agencies (p-values < 0.0001). The FDA was not different from the CDC, DOI and NOAA regarding the response proportions (p-values $\geq$ 0.19).

These findings may appear to be at odds with other responses that suggest scientists do not perceive their agency to maintain scientific integrity (e.g., the perception of leadership politicizing science used in decisionmaking). We believe, however, that responses to other quantitative survey questions, as well as to the open-ended questions of the survey, provide an answer to this conundrum. Federal scientists perceive the adherence to scientific integrity differently across groups of workers within their agency. Specifically, there is evidence to suggest federal scientists distinguish scientific integrity of career staff and those working immediately around them from those in politically appointed positions. Within their own community of career-level scientists, it seems that federal scientists perceive those individuals adhere to the policy. For example, respondents across agencies gave generally positive responses to the question, "My direct supervisor consistently stands behind scientists who put forth scientifically defensible positions that may be politically contentious." Specifically, 363 CDC respondents (63.9%), 830 DOI respondents (73.6%), 285 EPA respondents (64.9%), 222 FDA respondents (65.5%), and 765 NOAA respondents (67.4%) agreed or strongly agreed with this statement. As an anonymous respondent from the National Park Service responded, "From my perspective, the employees I work with have the highest standards of scientific integrity, remain focused on their work and clearly accept climate change as the primary issue in the future of our public lands." However, across many agencies respondents noted that they did not feel political appointees were adhering to the policy, as this USGS respondent wrote, "Scientific integrity within the Scientists at the USGS is impeccable. This does not seem to be the case for people in appointed leadership positions within the DOI." In its own survey on scientific integrity, the EPA also found that its staff's perception of adherence to the agency's scientific integrity policy varied depending on the group of individuals being referenced [58]. For example, a significant majority of EPA survey respondents (88%, 3,338 responses) reported that they would feel comfortable reporting allegations to supervisors, despite some respondents expressing concern about political interference and loss of scientific integrity by agency leadership. When asked about the culture of scientific integrity at the EPA in the three years prior to the survey, respondents to the EPA-conducted survey most prevalently reported positive experiences and

viewpoints but also expressed concern about agency management and leadership with respect to scientific integrity.

Another potential explanation for this discrepancy is the distinction between the provisions of scientific integrity policies and the scientific integrity violations observed. In many cases, federal scientists expressed concerns about the actions of appointed leadership that fall outside of the scope of agency scientific integrity policies, which tend to focus instead on the rights and responsibility of agency scientists and other career level staff. Therefore, it may be that federal agencies are adhering to the agency's scientific integrity policies, but federal scientists observed science-policy actions perceived as ethically problematic that fall outside of the scope of the policy.

## Response rate and potential bias

The survey response rate was notably lower than prior year surveys conducted using similar methods. Several factors may have contributed to this. Both active and passive discouragement from agency officials for taking the survey may have played a role. While leadership at several agencies, including the EPA and NOAA, encouraged employees to take the survey, some potential participants received communications from agency officials instructing them not to take the survey. The authors are aware of such communications being received by subsets of potential participants at the EPA, FWS, and NOAA. Relatedly, fear of retaliation for taking the survey from agency leadership, either with or without explicit instruction to not take the survey, may have contributed to lower response rates [59]. Individual open-ended responses to the survey suggested that some agency scientists perceived that either they or their colleagues would not feel comfortable filling out the survey and/or disclosing details in the current political environment.

Lower response rates compared to similar prior surveys may also be a function of greater concerns about phishing scams in recent years. The frequency and prominence of such scams where people fell victim by clicking links in emails from unknown addresses has increased greatly since the years when prior online surveys were conducted [39, 40].

Finally, survey fatigue may be partially responsible for lower response rates, as federal employees, particularly between 2016 and 2018, received a high level of outside inquiries from journalists and nongovernmental organizations; and high frequency of survey invitations has been shown to decrease response rates.

There is potential for bias in the sample of individuals who responded to the survey. Though the survey was administered via Iowa State University, level of familiarity and agreement with the Union of Concerned Scientists and its policy positions may have influenced who responded to the survey and the responses given. While the language of most survey questions was identical to that of prior surveys asked under different administrations, there is potential for response bias as is true for all survey work [60]. As the survey was sent to all potentially eligible participants, the sample was not random. Further, it is likely that many ineligible people received invitations to take the survey, particularly for agencies with no available job titles to target potential participants. For these reasons, survey results should not be considered representative of views of federal scientists at large; rather, survey results provide a window into perceptions that a select set of agency scientists hold and how such views vary across agencies and time.

Further, this is the first time to our knowledge that it has been attempted to assess scientist perceptions of scientific integrity at federal government agencies using quantitative statistical methods. Potential for validation of results was limited in this study design; however, internal results were found to be consistent, and this work can be considered a starting point for further work to quantitatively assess perceptions of scientific integrity at federal agencies.

## Conclusion

Our results indicate that federal scientists perceive losses of scientific integrity under the Trump Administration, given responses to key questions on the 2018 survey and comparison to surveys conducted prior to 2016. Respondent perceptions of scientific integrity appear to be influenced by their perceptions of the trustworthiness and competence of agency and administration leadership, the degree to which they feel valued and effective in the workplace, and the presence of clear and comprehensive scientific integrity policies and infrastructure. Differences in respondents' perceptions of scientific integrity between federal agencies can be explained by these three factors identified as contributing to government scientists' perceptions of agency scientific integrity.

The existence and implementation of a clear comprehensive scientific integrity policy at the agency or department level can explain some of the variability in responses observed. Prior to 2011, most government science agencies did not have scientific integrity policies in place. This lack of guidance, procedure, and infrastructure is apparent in surveys prior to 2011 where respondents across agencies indicated scientific integrity challenges in the workplace. At the time of the 2015 and 2018 surveys, the CDC, DOI, EPA, FDA, and NOAA all had comprehensive scientific integrity policies in place, as well as scientific integrity infrastructure including scientific integrity officers, trainings, and committees within agencies [37, 46, 61]. Such improvement in scientific integrity policies and processes at agencies is reflected in the 2018 survey results, where respondents reported generally high marks in response to question 10 on their agency's adherence to its scientific integrity policy.

Additionally, survey results indicate the significance of competent and trustworthy agency leadership as a contributing factor to federal scientists' perception of agency scientific integrity. At the time the survey was conducted, only NOAA didn't yet have a senate-confirmed head. The DOI, its agencies included in this analysis, the FDA, and the EPA all had Senate-confirmed political appointees leading the agency. Survey results, in addition to external evidence from independent reporting of agency policy actions, indicate that between 2016 and 2018, the EPA and DOI agencies experienced a higher degree of leadership changes, new policy directions, and divergence from business as usual on science policy issues compared to the FDA and NOAA. This could explain why respondents at the FDA and NOAA reported generally more positive results that were significantly different from those at the DOI and EPA.

Finally, survey results suggest that agency scientists' perceptions of scientific integrity may be influenced by the degree to which employees feel effective and valued in their jobs. At the DOI and the EPA, respondents reported higher degrees of censorship and self-censorship, notably around climate change related work. Such restrictions on production or communication of scientific work affect government scientists' ability to be effective at their job and influence the degree to which they feel their role is valued. Respondents at DOI agencies and the EPA also reported higher proportions of resource allocations away from politically contentious topics and being excluded from conversations they would normally be a part of than respondents from the FDA and NOAA—measures that also inform the degree to which employees feel valued and effective at work. These factors suggest that elements of the work environment at DOI agencies and the EPA may be contributing to scientists feeling less valued and effective at these agencies.

Overall, these results indicate that the presence of agency-wide scientific integrity policies and infrastructure alone do not predict government scientists' perceptions of their agency's degree of effectiveness or adherence to those policies. There are a variety of nuances under the umbrella of scientific integrity. This suggests that at the time the survey was conducted, agency scientific integrity policies may either not have been put into effective practice at all levels of

agencies or may not be sufficiently comprehensive to ensure agency scientists perceive scientific integrity to be a high priority in their workplace. Results suggest that perceptions of competent and trustworthy leadership and feeling effective, well-resourced, and valued at work seem to be contributing factors. These findings suggest that enhancing scientific integrity at federal science agencies may necessitate competent and trustworthy leadership, a positive work environment for federal scientists and comprehensive scientific integrity policies and infrastructure.

Future work should consider ways of increasing response rates in order to increase the number, variety, and representativeness of perspectives collected across federal agencies. Additional surveys or incorporation of questions related to scientific integrity into scheduled surveys of federal agencies would provide comparative data to further assess government scientists' perceptions of the multiple components that together make up scientific integrity.

## Supporting information

**S1 Appendix. Survey of federal scientists response rates, 2018.**
(DOCX)

**S2 Appendix. Survey E-mail invitation.**
(DOCX)

**S3 Appendix. 2018 survey of federal scientists (paper version).**
(DOCX)

**S4 Appendix. Quantitative survey results.**
(PDF)

**S5 Appendix. Open-ended survey results.**
(XLSX)

**S6 Appendix. Quantitative statistical assessment code.**
(ZIP)

## Acknowledgments

The authors would like to thank those who put in lots of time and effort in developing the survey instrument, organizing and analyzing vast amounts of survey data, and providing feedback on earlier versions of this manuscript including Charise Johnson, Anita Desikan, Emily Berman, Andrew A. Rosenberg, Michael Halpern, Liz Borkowski, and Susan Wood. We'd also like to thank the staff at the Center for Survey Statistics and Methodology at Iowa State University for their help in implementing the survey to thousands of federal scientists. Lastly, we thank the federal scientists themselves for participating in this survey and for the work they do every day.

## Author Contributions

**Conceptualization:** Gretchen T. Goldman, Jacob M. Carter.

**Data curation:** Yun Wang, Janice M. Larson.

**Formal analysis:** Gretchen T. Goldman, Jacob M. Carter, Yun Wang, Janice M. Larson.

**Investigation:** Gretchen T. Goldman, Jacob M. Carter.

**Methodology:** Gretchen T. Goldman, Jacob M. Carter, Yun Wang, Janice M. Larson.

**Project administration:** Gretchen T. Goldman, Jacob M. Carter, Yun Wang, Janice M. Larson.

**Resources:** Janice M. Larson.

**Software:** Yun Wang.

**Supervision:** Gretchen T. Goldman, Janice M. Larson.

**Validation:** Gretchen T. Goldman, Jacob M. Carter, Yun Wang, Janice M. Larson.

**Visualization:** Jacob M. Carter, Yun Wang.

**Writing – original draft:** Gretchen T. Goldman, Jacob M. Carter.

**Writing – review & editing:** Yun Wang, Janice M. Larson.

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
