## [Decision Letter · Decision Letter 0]

9 Jan 2020

PONE-D-19-31764

Perceived Losses of Scientific Integrity under the Trump Administration: A Survey of Federal Scientists

PLOS ONE

Dear Dr Goldman,

Thank you for submitting your manuscript to PLOS ONE. After careful consideration, we feel that it has merit but does not fully meet PLOS ONE’s publication criteria as it currently stands. Therefore, we invite you to submit a revised version of the manuscript that addresses the points raised during the review process.

You are expected to revise the article according to the reviewers' comments and provide answers to their questions, very carefully. After the resubmission of your article, the journal's editorial office may invite more reviewers to ensure of the quality of the presentation of your findings before publication. 

We would appreciate receiving your revised manuscript by Feb 23 2020 11:59PM. To enhance the reproducibility of your results, we recommend that if applicable you deposit your laboratory protocols in protocols.io, where a protocol can be assigned its own identifier (DOI) such that it can be cited independently in the future. For instructions see: http://journals.plos.org/plosone/s/submission-guidelines#loc-laboratory-protocols

We look forward to receiving your revised manuscript.

Kind regards,

Prof, Mojtaba Vaismoradi, PhD, MScN, BScN

Academic Editor

PLOS ONE

Journal Requirements:

'This research was funded by supporters of the Union of Concerned Scientists.'

'The authors received no specific funding for this work.'

Reviewers' comments:

Reviewer #1: Although finding of this study are interesting, the method section is not clear and many concerns are existing regarding the methodology:

There was no logic behind sample size calculation. The method of sampling was not clear and it needs a deep revision. In my opinion, the way of data collection is biased as many invited individuals refused to answer email. Also, by using this design any judgment about the study findings is misleading. More importantly, for data analysis too simplistic method (X2 test) was used which prevent us to have any decision about findings because this is a descriptive study with a cross sectional design.

I am not sure if the questionnaire was used for this research has been validated before.

How authors reached 3 factors for Factor Contributing to Scientific Integrity Perceptions? Did they used factor analysis?!

Reviewer #2: The manuscript is technically sound. Its conclusions are supported by the data it presents. The authors provide an extensive discussion of several important methodological issues including sampling, survey administration, and data analysis. However, more details on the validity of the survey measures would be welcome. The manuscript mentions that most are replicated from earlier surveys, which is a good sign. However, the internally inconsistent survey results discussed on pp 26-27 suggest some murkiness in what these items actually measure. The open ended survey responses provide some preliminary clues to explain this discrepancy, although more details on the validity of the survey measures would strengthen the paper. Also, I recommend reproducing the figures in grayscale so that they are legible when printed in black and white.

The presentation of percentages in the results section is an appropriate and effective way to interpret survey findings. However, I am confused by the various chi-square tests, which assume random sampling. While the novelty of the survey results warrants publication even in the absence of hypothesis testing, respondents are certainly not a random sample of the population of government scientists. The authors should address this issue to help readers make better sense of the statistical inferences that can be drawn from these very interesting data.

The survey instrument and other supporting materials are available and so are some aggregate survey statistics. However, the raw data and syntax files are not included in the supporting information.

The manuscript is well written and presented at a level that is appropriate for scientific publication.

---

## [Author Response · Author response to Decision Letter 0]

22 Feb 2020

Reviewers' comments:

Reviewer #1: 

Although finding of this study are interesting, the method section is not clear and many concerns are existing regarding the methodology: There was no logic behind sample size calculation. The method of sampling was not clear and it needs a deep revision. In my opinion, the way of data collection is biased as many invited individuals refused to answer email. Also, by using this design any judgment about the study findings is misleading. More importantly, for data analysis too simplistic method (X2 test) was used which prevent us to have any decision about findings because this is a descriptive study with a cross sectional design.

Response:

We acknowledge the limitations of the sampling method and resulting limitation in how the data can be interpreted. This is acknowledged in the Response Rate and Potential Bias section of the Results and Discussion (lines 728-738 in the revised manuscript with tracked changes). Additionally, in response to this comment, we have added a sentence to the Results and Discussion section explicitly acknowledging that the sample is not random (lines 733-734), and we have added language to make this point clearer under the Survey Administration section of the Materials and Methods (lines 259-261). 

With regard to the choice of statistical test, we note that the Chi-square test is a non-parametric statistic that assumes the data were obtained through random selection from a defined sample, as the review states. However, it is a similarly useful tool for assessing associations and making inferences when data are from non-probability samples. The difference is in the application and analysis of the results. If the sample is probability-based, the results of Chi-square test results are representative of the population as a whole. If Chi-square tests are used with data obtained from a non-probability sample, the results are representative of the respondents but not necessarily representative of the larger population. This is why we have made clear that these results are not to be interpreted as representative of all federal scientists, but rather a window into the perceptions of a subset of them. That’s why we have added text in the Materials and Methods and Results and Discussion sections to make this clearer as noted above.

Reviewer #1:

I am not sure if the questionnaire was used for this research has been validated before.

How authors reached 3 factors for Factor Contributing to Scientific Integrity Perceptions? Did they used factor analysis?!

Response:

We appreciate the reviewer’s comments on validation. To our knowledge, this is the first time that it has been attempted to assess scientist perceptions of scientific integrity at federal government agencies using quantitative statistical methods. Potential for validation of results was limited in this study design; however, internal results were found to be consistent, and this work can be considered a starting point for further work to quantitatively assess perceptions of scientific integrity at federal agencies. 

Since we believe that the survey factors contributing to scientific integrity perceptions are not exhaustive, factor analyses might not be the concern to address. But the internal consistency of these items can be and were assessed by using Cronbach’s alpha. Below are the results of such an analysis.

Cronbach’s alpha is not a statistical test; it is a coefficient of reliability (or consistency). High value of Cronbach’s alpha does not imply these items are unidimensional (only one factor behind the direct measurements). 

Some values were widely applied to use Cronbach’s alpha. 

Cronbach's alpha Internal consistency

0.9 ≤ α Excellent

0.8 ≤ α < 0.9 Good

0.7 ≤ α < 0.8 Acceptable

0.6 ≤ α < 0.7 Questionable

0.5 ≤ α < 0.6 Poor

α < 0.5 Unacceptable

Here is the summary of Cronbach’s alpha for survey items that contribute to scientific integrity perceptions (Note: The below analysis is better viewed in the Response to Reviewers file. The overall raw Cronbach’s alpha is around 0.52 without considering question 1 items, which are different from the others. For the other 9 items, question 10 is not very correlated with others, and the Cronbach’s alpha increases without question 10. 

Cronbach's Alpha Assessment for All Agencies

 all 10 items Q10 other 9 items

all answers [1] <0 <10 0.528

 all 9 items remove Q10 remove Q9 remove Q9, Q10 Q2 

Q3 

 Q4 Q5 Q6 Q7 Q8

answers used in analyses [2] 0.5 0.6 0.66 0.76 0.79 0.617

answers in ordinal scale [3] 0.45 0.61 0.64 0.778 0.79 0.642

Simplified answers [4] 0.408 0.554 0.592 0.731 0.74 0.582

Cronbach's Alphas Assessment for Select Agencies

 all 10 items Q1 other 9 items

all answers [1] <0 <10 0.524

 all 9 items remove Q10 remove Q9 remove Q9, Q10 Q2 

Q3 

 Q4 Q5 Q6 Q7 Q8

answers used in analyses [2] 0.524 0.607 0.667 0.757 0.788 0.62

answers in ordinal scale [3] 0.524 0.607 0.667 0.757 0.788 0.62

Simplified answers [4] 0.406 0.555 0.589 0.73 0.746 0.583

[1] all possible answers provided in survey including ‘prefer not to disclose’

[2] Answers used in general analyses including “don’t know”

[3] Answers used in ordinal analyses excluding “don’t know”

[4] Final simplified 3-level answers: negative, positive and neutral

The selected agencies were determined by the response rates and their answers are almost identical with all agencies combined. Without questions 9 and 10 series items, the Cronbach’s alpha is larger than 0.7, the acceptable consistency. The question with two components (questions 2 and 3) was consistent, questions 4-8 were questionably consistent. Given the study design and interpretation of results presented in this work, we believe this assessment adequately demonstrates the reliability of the results. 

Per this reviewer comment and analysis, we have added text to the Results and Discussion section (lines 739-743) to discuss validity, consistency, and limitations of the results.

Reviewer #2: 

The manuscript is technically sound. Its conclusions are supported by the data it presents. The authors provide an extensive discussion of several important methodological issues including sampling, survey administration, and data analysis. However, more details on the validity of the survey measures would be welcome. The manuscript mentions that most are replicated from earlier surveys, which is a good sign. However, the internally inconsistent survey results discussed on pp 26-27 suggest some murkiness in what these items actually measure. The open-ended survey responses provide some preliminary clues to explain this discrepancy, although more details on the validity of the survey measures would strengthen the paper. 

In addition to the above text changes and discussion on internal consistency using Cronbach’s alpha assessment in response to concerns of Review #1, we have added additional evidence and context to the consistency discussion of the Results and Discussion (lines 670-697), including additional quantitative survey response data that support the proposed explanation for the consistency (lines 677-682), as well as details about the separately produced survey conducted by the EPA that also had results that aligned with our results and explanation (lines 691-697). 

Reviewer #2:

Also, I recommend reproducing the figures in grayscale so that they are legible when printed in black and white.

Response:

We are open to working with the editorial team to produce figures that are most accessible to readers. 

The presentation of percentages in the results section is an appropriate and effective way to interpret survey findings. However, I am confused by the various chi-square tests, which assume random sampling. While the novelty of the survey results warrants publication even in the absence of hypothesis testing, respondents are certainly not a random sample of the population of government scientists. The authors should address this issue to help readers make better sense of the statistical inferences that can be drawn from these very interesting data.

As described in above response to Review #1 comments, we acknowledge the limitations of the survey methods and have made edits to further clarify these limitations in the Methods and Results & Discussion sections. 

The survey instrument and other supporting materials are available and so are some aggregate survey statistics. However, the raw data and syntax files are not included in the supporting information.

We have added raw data tables (Appendix D), as well as the code use to conduct the analysis to the Supporting Information. 

Reviewer #2:

The manuscript is well written and presented at a level that is appropriate for scientific publication.

---

## [Decision Letter · Decision Letter 1]

16 Mar 2020

PONE-D-19-31764R1

Perceived Losses of Scientific Integrity under the Trump Administration: A Survey of Federal Scientists

PLOS ONE

Dear Dr Goldman,

Thank you for submitting your manuscript to PLOS ONE. After careful consideration, we feel that it has merit but does not fully meet PLOS ONE’s publication criteria as it currently stands. Therefore, we invite you to submit a revised version of the manuscript that addresses the points raised during the review process.

We would appreciate receiving your revised manuscript by Apr 30 2020 11:59PM. To enhance the reproducibility of your results, we recommend that if applicable you deposit your laboratory protocols in protocols.io, where a protocol can be assigned its own identifier (DOI) such that it can be cited independently in the future. For instructions see: http://journals.plos.org/plosone/s/submission-guidelines#loc-laboratory-protocols

We look forward to receiving your revised manuscript.

Kind regards,

Prof, Mojtaba Vaismoradi, PhD, MScN, BScN

Academic Editor

PLOS ONE

Editor comment:

Please do a  literature search and provide some descriptions of previous similar studies in the introduction section. That would be fine for your article readers if they find that such a topic has been considerd by previous researchers  even in other disciplines. Also, it improves the justification for the significance of your study. 

---

## [Author Response · Author response to Decision Letter 1]

1 Apr 2020

Editor comments:

Please do a literature search and provide some descriptions of previous similar studies in the introduction section. That would be fine for your article readers if they find that such a topic has been considered by previous researchers even in other disciplines. Also, it improves the justification for the significance of your study.

Response: 

We have added discussion of published literature relevant to this work, including previous surveys of scientific experts, studies linking job satisfaction and respect for managers with fairness and ethics, and other relevant studies assessing job satisfaction survey results [lines 110-142 on the revised manuscript with tracked changes].

Additionally, we have modified a few sentences of the Results & discussion section [lines 509-528]. These edits are in response to a comment from reviewer #2 on the previous round of review in which the reviewer requested discussion of what survey items “actually measure.” On this survey item (question 4), we had omitted CDC results; these have been added in and some qualitative survey response examples have been added to help readers interpret what this item is measuring, highlighting budget concerns as an example of how inadequate financial resources influences respondents’ views of agency effectiveness. 

Finally, we have added supporting information that includes the collection of open-ended survey responses, from which some manuscript text draws when qualitative results are discussed, and we have organized the Supporting Information Files into ordered appendices for clarity.

---

## [Editor Report · Decision Letter 2]

6 Apr 2020

Perceived Losses of Scientific Integrity under the Trump Administration: A Survey of Federal Scientists

PONE-D-19-31764R2

Dear Dr. Goldman,

We are pleased to inform you that your manuscript has been judged scientifically suitable for publication and will be formally accepted for publication once it complies with all outstanding technical requirements.

With kind regards,

Prof, Mojtaba Vaismoradi, PhD, MScN, BScN

Academic Editor

PLOS ONE

---

## [Editor Report · Acceptance letter]

9 Apr 2020

PONE-D-19-31764R2 

Perceived Losses of Scientific Integrity under the Trump Administration: A Survey of Federal Scientists 

Dear Dr. Goldman:

I am pleased to inform you that your manuscript has been deemed suitable for publication in PLOS ONE. Congratulations! Your manuscript is now with our production department. 

With kind regards,

on behalf of

Professor Mojtaba Vaismoradi 

Academic Editor

PLOS ONE